# One-Dimensional Mercury Halide Coordination Polymers Based on A Semi-Rigid *N*-Donor Ligand: Reversible Structural Transformation

**DOI:** 10.3390/polym11030436

**Published:** 2019-03-06

**Authors:** Pradhumna Mahat Chhetri, Xiang-Kai Yang, Chih-Tung Yang, Jhy-Der Chen

**Affiliations:** 1Department of Chemistry, Chung Yuan Christian University, Chung-Li 320, Taiwan; mahatp@gmail.com (P.M.C.); XiangKai-Yang@outlook.com (X.-K.Y.); gm7229358@gmail.com (C.-T.Y.); 2Department of Chemistry, Amrit Science Campus, Tribhuvan University, Kathmandu 44600, Nepal

**Keywords:** coordination polymer, bis-pyridyl-bis-amide, structural transformation, mercury halide

## Abstract

Four one-dimensional (1D) mercury(II) halide coordination polymers have been synthesized by using a semi-rigid *N*-donor ligand, 2,2′-(1,4-phenylene)-bis(*N*-(pyridin-3-yl)acetamide) (1,4-pbpa). While [Hg(1,4-pbpa)Cl_2_·CH_3_OH]_n_, **1**, forms a sinusoidal chain, the complexes [Hg(1,4-pbpa)X_2_]_n_ (X = Cl, **2**; Br, **3**; I, **4**) are helical. The sinusoidal **1** undergoes reversible structural transformation with helical **2** upon removal and uptake of CH_3_OH, which was accompanied with the conformation adjustment of the 1,4-pbpa ligand from *trans* anti-anti to *trans* syn-anti. Pyridyl ring rotation of the 1,4-pbpa ligand that results in the change of the ligand conformation is proposed for the initiation of the structural transformation.

## 1. Introduction

The rational design and synthesis of novel coordination polymers (CPs) not only afford interesting structural topologies but also extend the range of potential applications in magnetism, luminescence, catalysis, and gas storage [1]. The preparation methods, metal-ligand ratio, and solvent combination play key roles in determining the structural diversity and properties, which can also be observed by the influence of the spacer ligands and metal identities. The structural transformations are intriguing in CPs due to their potential applications in switches and sensors, which can be triggered by exchange of guest molecules, removal and uptake of solvents and external stimuli like heat, light, and mechanical forces [2]. The synthesis and structures of a variety of one-dimensional (1D) mercury(II) CPs have been reported, however, it remains a challenge to elucidate their structure-ligand relationship and thereby the intrinsic properties [3,4,5,6,7,8,9,10,11,12,13,14].

We have made our effort to investigate the structural diversity and properties of the CPs constructed from the bis-pyridyl-bis-amide ligands [10] and several Hg(II) CPs that exhibited structural transformations have been reported [4,5,11]. The HgI_2_-containing CP [Hg(1,2-pbpa)I_2_]_n_ [1,2-pbpa = 2,2′-(1,2-phenylene)-bis(*N*-pyridin-3-yl)acetamide] has been shown to display reversible structural transformation with [Hg(1,2-pbpa)I_2_·MeOH]_n_ and [Hg(1,2-pbpa)I_2_·MeCN]_n_ upon adsorption/desorption and exchange of methanol and acetonitrile molecules, which demonstrate the importance of N–H---X (X = halide anion) and Hg---X interactions in the evaluation of structural transformation [4]. Moreover, reversible structural transformations between [Hg(1,3-pbpa)X_2_]_n_ [X = Br and I; 1,3-pbpa = 2,2′-(1,3-phenylene)-bis(*N*-(pyridin-3-yl)acetamide] and [Hg(1,3-pbpa)X_2_·MeCN]_n_ were ascribed to the formation and breaking of the N–H---N hydrogen bonds to the acetonitrile molecules [5]. Although several HgCl_2_ CPs containing the bis-pyridyl-bis-amide ligands have been prepared [4,5,13,14,15], none of them has been reported to show structural transformation upon removal and uptake of solvent. The HgCl_2_-containing CPs constructed from 2-(2-hydroxyethyl)pyridine have been found to proceed reversible 1D-2D structural transformation through multiple covalent bond breaking and bond reforming [11].

To investigate the effect of ligand-isomerism on the formation mercury(II) halide-containing CPs as well as the impact on the properties, we have carried out the reactions of 2,2′-(1,4-phenylene)-bis(*N*-(pyridin-3-yl)acetamide (1,4-pbpa) with the Hg(II) halide salts. Herein, we report the syntheses, structures and emission properties of the 1D CPs [Hg(1,4-pbpa)Cl_2_·CH_3_OH]_n_, 1, and [Hg(1,4-pbpa)X_2_]_n_ (X = Cl, 2; Br, 3; I, 4). Reversible structural transformation between sinusoidal 1 with the *trans* anti-anti ligand conformation and helical 2 with the *trans* syn-anti conformation upon removal and uptake of CH_3_OH is reported. The factors that govern the reversible structural transformation are also discussed.

## 2. Experimental Section

### 2.1. General Procedures

The elemental analyses (C, H, and N) were performed using a PE 2400 series II CHNS/O (PerkinElmer Instruments, Shelton, CT, USA) or an Elementar Vario EL-III analyzer (Elementar Analysensysteme GmbH, Hanau, German). The IR spectra (KBr disk) were recorded on a JASCO FT/IR-460 plus spectrometer ((JASCO, Easton, MD, USA). The powder X-ray diffraction was carried out by using a Bruker D2 PHASER diffractometer (Bruker Corporation, Karlsruhe, Germany) equipped with a CuK_α_ (λ = 1.54 Å) radiation. The solid state emission spectroscopy was done using a Hitachi F-4500 spectrometer (Hitachi, Tokyo, Japan) with excitation slit = 5.0 nm and emission slit = 5.0 nm at room temperature.

### 2.2. Materials

The reagents 1,4-phenylenediacetic acid was purchased from ACROS Co. (Pittsburgh, PA, USA), and 3-aminopyridine, pyridine, tripenylphosphite, and mercury(II) halide salts from Alfa Aesar Co. (Heysham, UK). The ligand 2,2′-(1,4-phenylene)-bis(*N*-(pyridin-3-yl)acetamide) (1,4-pbpa) was prepared according to a published procedure [16].

### 2.3. Preparations

#### 2.3.1. [Hg(1,4-pbpa)Cl_2_·CH_3_OH]_n_, 1

A 20 mL MeOH solution of HgCl_2_ (0.054 g, 0.20 mmol) was layered on top of a 20 mL THF solution of 1,4-pbpa (0.069 g, 0.20 mmol). After two weeks, colorless crystals of **1** were generated, which were then collected. Yield: 0.032 g (25%). Anal. calcd for C_21_H_22_Cl_2_HgN_4_O_3_ (*M_W_* = 649.91): C, 38.81; N, 8.62; H, 3.41%. Found: C, 38.48; N, 8.94; H, 2.91%. IR (cm^−1^): 560(w), 639(w), 701(m), 735(w), 781(w), 812(m), 968(w), 1026(w), 1047(w), 1105(w), 1148(m), 1196(m), 1244(w), 1300(m), 1345(s), 1427(s), 1478(s), 1540(s), 1593(m), 1675(s, C=O), 2918(w), 3030(w), 3063(w), 3131(w), 3190(w), 3303(w, N–H stretch).

#### 2.3.2. [Hg(1,4-pbpa)Cl_2_]_n_, 2

Compound **2** was prepared by following the procedure described for **1**, except a 20 mL EtOH solution of HgCl_2_ was used. Yield: 0.030 g (20%). Anal. calcd for C_20_H_18_Cl_2_HgN_4_O_2_ (*M_W_* = 617.87): C, 38.88; N, 9.07; H, 2.94%. Found: C, 38.95; N, 9.03; H, 2.85%. IR (cm^−1^): 416(w), 504(m), 557(m), 592(m), 636(m), 701(s), 739(m), 782(m), 811(s), 864(w), 923(w), 961(w), 1023(w), 1051(m), 1108(m), 1149(s), 1204(m), 1248(m), 1274(s), 1299(m), 1353(s), 1428(s), 1480(s), 1548(s), 1587(s), 1673(s, C=O), 2921(w), 3075(w), 3134(m), 3269(s, N–H stretch), 3358(s, N–H stretch).

#### 2.3.3. [Hg(1,4-pbpa)Br_2_]_n_, 3

A 20 mL MeOH or EtOH solution of HgBr_2_ (0.072 g, 0.20 mmol) was layered on top of a 20 mL THF solution of 1,4-pbpa (0.069 g, 0.20 mmol). After two weeks, colorless crystals of **3** were generated, which were then collected. Yield: 0.047 g (33%). Anal. calcd for C_20_H_18_Br_2_HgN_4_O_2_ (*M_W_* = 706.79): C, 33.99; N, 7.93; H, 2.57%. Found: C, 34.36; N, 7.79; H, 2.22%. IR (cm^−1^): 416(w), 504(m), 557(m), 592(m), 636(m), 701(s), 739(m), 782(m), 811(s), 864(w), 923(w), 961(w), 1023(w), 1051(m), 1108(m), 1149(s), 1204(m), 1248(m), 1274(s), 1299(m), 1344(s), 1428(s), 1480(s), 1546(s), 1586(s), 1672(s, C=O), 2921(w), 3075(w), 3134(m) , 3262(s, N–H stretch), 3358(s, N–H stretch).

#### 2.3.4. [Hg(1,4-pbpa)I_2_]_n_, 4

Compound **4** was prepared by following the procedure described for **3**, except HgI_2_ (0.091 g, 0.20 mmol) was used. Yield: 0.061 g (38%). Anal. calcd for C_20_H_18_HgI_2_N_4_O_2_ (*M_W_* = 800.77): C, 29.99; N, 6.99; H, 2.27%. Found: C, 30.46; N, 6.95; H, 2.4%. IR (cm^−1^): 416(w), 504(m), 557(m), 592(m), 636(m), 701(s), 739(m), 782(m), 811(s), 864(w), 923(w), 961(w), 1023(w), 1051(m), 1108(m), 1149(s), 1204(m), 1248(m), 1274(s), 1299(m), 1342(s), 1427(s), 1478(s), 1546(s), 1586(s), 1670(s, C=O), 2921(w), 3075(w), 3134(m) , 3250(s, N–H stretch), 3355(s, N–H stretch).

The purities of complexes **1**–**4** have been examined by using the PXRD patterns, Appendix A.

### 2.4. X-ray Crystallography

A Bruker AXS SMART APEX II CCD diffractometer (Bruker AXS, Madison, WI, USA) that was equipped with a graphite monochromated MoKα (λ = 0.71073 Å) radiation and operated at 50 kV and 30 mA was used to collect the diffraction data for complexes **1**–**4**. Data reduction was performed by using well-established computational procedures with empirical absorption correction based on “multi-scan” [17]. Patterson or direct method was used to locate the positions of some of the heavier atoms and the remaining atoms were found in a series of alternating difference Fourier maps and least-square refinements, while hydrogen atoms were added by using the HADD command in SHELXTL [18]. The CH_3_OH molecule of **1** is disordered such that two sets of orientations of the oxygen atom can be shown. Basic crystal parameters and structure refinement are summarized in Table 1.

## 3. Results and Discussion

### 3.1. Synthesis

Complexes **1**–**4** were prepared by layering the MeOH or EtOH solution of mercury(II) halide on top of the THF solution of 1,4-pbpa. Using the combination of MeOH and THF solvents, the reaction of HgCl_2_ with 1,4-pbpa afforded the sinusoidal **1** containing the co-crystallized MeOH molecule, while that of EtOH and THF gave helical **2** without solvent co-crystallization. However, either in MeOH or EtOH, reactions of HgBr_2_ and HgI_2_ with 1,4-pbpa gave **3** and **4**, respectively, without solvent co-crystallization. The powder patterns of helical **3** and **4** prepared from HgBr_2_ and HgI_2_ in MeOH/THF and EtOH/THF are shown in Appendix A, respectively, which indicate that formation of **3** and **4** is irrespective of the solvent system. Scheme 1 depicts the synthetic pathways for **1**–**4**.

### 3.2. Structural Descriptions

#### 3.2.1. Structure of **1**

Single-crystal X-ray structural analysis shows that the crystals of **1** conform to the monoclinic space group *P*2_1_/*m* with a half of Hg(II) ion, a half of 1,4-pbpa ligand, two halves of chloride anion and a half of methanol molecule in an asymmetric unit. Figure 1a depicts a drawing showing the coordination environment of metal ion, which is coordinated with two pyridyl nitrogen atoms [Hg-N = 2.432(3) Å] from two 1,4-pbpa ligands and two chloride anions [Hg–Cl = 2.3448(19) and 2.3700(18) Å], resulting in a distorted tetrahedral geometry (τ_4_ = 0.75) [19,20]. The 1,4-pbpa ligands bridge the adjacent metal ions to form a 1D sinusoidal chain, Figure 1b. The repeating unit reveals the period of 20.66 Å involving two 1,4-pbpa ligands and two Hg(II) ions, which is slightly less than the Hg---Hg distance of 20.82 Å separated by the 1,4-pbpa ligand, while the amplitude of the sinusoidal chain is 9.04 Å. The dihedral angle of pyridine-pyridine ring is 0° and that of benzene-pyridine ring is 69.44°. Two-dimensional supramolecular structure is stabilized by N–H---O [H---O = 2.16 Å; N---O = 2.96 Å; ∠N–H---O = 153.2°] hydrogen bonding originating from the amide hydrogen atoms to the amide oxygen atoms in the adjacent chains, as shown in Figure 1c. Figure 1d reveals that these interlinked chains stacked along the *a* axis and the methanol molecules are located in the concave-convex sites, which interact with the methylene hydrogen atoms of 1,4-pbpa ligands through weak C–H---O hydrogen bonds (H---O = 3.14 Å; C---O = 3.98 Å; ∠C–H---O = 146.6°) [21].

#### 3.2.2. Structures of **2**–**4**

Single crystal X-ray structural analyses revealed that isomorphous crystals of **2**–**4** conform to the monoclinic space group *P*2_1_/*c* with one Hg(II) ion, one 1,4-pbpa ligand and two halide anions in an asymmetric unit. Figure 2a depicts a representative drawing showing the coordination environment of the metal ion, which is coordinated with two halide ions [Hg–X = 2.3552(1) and 2.3581(9) Å for **2**; 2.4805(5) and 2.4846(5) Å for **3**; 2.6488(5) and 2.6462(5) Å for **4**] and two pyridyl nitrogen atoms [Hg–N = 2.415(2) and 2.452(2) Å for **2**; 2.404(3) and 2.455(3) Å for **3**; 2.416(5) and 2.477(5) Å for **4**] from two 1,4-pbpa ligands, resulting in a distorted tetrahedral geometry (τ_4_ = 0.74, **2**; 0.75, **3**; 0.77, **4**). Complexes **2**–**4** form 1D helical chains having the period of 16.29, 16.38 and 16.57 Å, respectively, involving one 1,4-pbpa ligand and two Hg(II) cations, Figure 2b. The 1D helical chains are supported by the N–H---O [H---O = 2.11 Å; N---O = 2.86 Å; ∠N–H---O = 144.3^o^, **2**; H---O = 2.10 Å; N---O = 2.83 Å; ∠N–H---O = 142.2°, **3**; [H---O = 2.07 Å; N---O = 2.80 Å; ∠N–H---O = 142.2°, **4**] hydrogen bonds originating from the amide groups in 1,4-pbpa and forming 2D supramolecular structures in a crossover fashion, as shown in Figure 2c.

#### 3.2.3. Ligand Conformation

The conformation of bis-pyridine-bis-amide has been determined by the orientations of C=O or N–H pairs and the orientations of the pyridyl nitrogen and amide oxygen. If the orientation of C=O or N–H pair are on the same direction, it is defined as “cis”, and if the pair is on the opposite direction, it is defined as “trans”. By considering the orientations of pyridyl nitrogen and amide oxygen, syn-syn, anti-anti and syn-anti are also specified [10,22,23]. Accordingly, the conformations of 1,4-pbpa are trans anti-anti in **1**, Scheme 2a, and trans syn-anti in **2**–**4**, Scheme 2b, indicating that the co-crystallized solvents play important role in determining the ligand conformation and, thus, the structural type. The detail of dihedral angles and conformations of 1,4-pbpa in **1**–**4** are listed in Table 2.

#### 3.2.4. Structural Comparisons

The structural difference between sinusoidal **1** and helical **2** implies the influence of the cocrystallized MeOH solvent molecules on the structural diversity of the mercury(II) chloride CPs constructed from 1,4-pbpa ligands. A comparison of the structures of **2**–**4** indicates that the identity of the halide anions shows no effect on the structural type of the Hg(II) halide CPs based on 1,4-pbpa.

It is worthwhile to investigate the isomeric effect of 1,2-, 1,3-, and 1,4-pbpa ligands on the structural diversity of the Hg(II) halide CPs, Table 3. Solvothermal reactions of HgX_2_ with 1,2-pbpa in ethanol afforded the isostructural 1D zigzag chains [Hg(1,2-pbpa)X_2_]_n_ (X = Cl, Br and I), while layering reactions of a ethanolic solution of 1,2-pbpa with a methanolic solution and an acetonitrile solution of HgI_2_ gave 1D helical chains [Hg(1,2-pbpa)I_2_·MeOH]_n_ and [Hg(1,2-pbpa)I_2_·MeCN]_n_, respectively [4]. On the other hand, solvothermal reactions of HgX_2_ salts with 1,3-pbpa in acetonitrile afforded the 1D helical chains [Hg(1,3-pbpa)X_2_]_n_ (X = Cl, Br and I), and the 1D mesohelical chains [Hg(1,3-pbpa)X_2_·MeCN]_n_ (X = Br and I) were obtained by layering solutions of HgX_2_ and 1,3-pbpa at room temperature [5]. For those complexes without co-crystallized solvents, the 1,3-pbpa and 1,4-pbpa ligands direct the same structural type of helical chain, which are in marked contrast to the 1,2-pbpa ligands that result in the zigzag chain. Moreover, structural variations are observed upon solvent co-crystallization, leading to the formation of helical chains, mesohelical chains, and sinusoidal chain for the Hg(II) CPs constructed from, 1,2-pbpa, 1,3-pbpa and 1,4-pbpa, respectively. The structural diversity of these pbpa-based Hg(II) halide CPs are thus most probably governed by the co-crystallized solvent molecules and ligand-isomerism of the pbpa ligands, whereas the role of the halide anions is not influential.

#### 3.2.5. Structural Transformation

The core structures of complexes **1** and **2** consist of the same formula and thus can be regarded as a pair of supramolecular isomers with and without the co-crystallized solvent molecules, which provide a unique opportunity for the study of the structural transformation between these two complexes upon removal and uptake of CH_3_OH. To investigate the structural transformation, we first checked feasibility of the structural change from **1** to **2** by heating **1** at variable temperatures, which was verified by using powder X-ray diffraction (PXRD) patterns. When complex **1** was heated from 50 to 100 °C, the PXRD patterns are significantly different from that of the simulation of **1**, indicating structural transformation. Heating **1** at 120 and 150 °C, respectively, afforded PXRD patterns that are comparable to that of the simulation of **2** except some peaks around 2θ = 7.5 and 16°. Moreover, increasing the temperature to 180 °C gave a PXRD pattern well matched with the simulated pattern of **2**, Figure 3. On the other hand, immersion of **2** into MeOH for one month showed no significant change on the PXRD pattern, Appendix A. However, when **2** was heated solvothermally at 120 °C, the pattern changed and well matched with that of the simulation of **1**, Figure 4. To the best of our knowledge, the reversible structural transformation between **1** and **2** represents the first example of the bis-pyridyl-bis-amide-based HgCl_2_ CPs. Attempts to investigate the structural transformation by immersing **3** and **4** into MeOH, as well as heating solvothermally, led no significant change on their PXRD patterns, Appendix A.

To propose the possible mechanism for the structural transformation between **1** and **2**, we consider the change in ligand conformation. When the sinusoidal **1** was heated to remove the CH_3_OH molecules, cleavage of the N–H---O hydrogen bonds between the stacking chains, Figure 1c, and formation of the N–H---O hydrogen bonds between the crossover chains, Figure 2c, result in the rearrangement of the ligand conformation from *trans* anti-anti to *trans* syn-anti and the formation of helical **2**, and vice versa, when **2** was solvothermally heated in CH_3_OH, the structure of **1** was recovered. Removal of the CH_3_OH from **1** thus triggers the reorientation of one of the pyridyl rings from “anti” to “syn”, leading to the conversion of the conformation *trans* anti-anti to *trans* syn-anti, Scheme 3. The rotation of the pyridyl rings drastically affects the Py-Py dihedral angles from 0 to 48.4° and C–C–C–O torsion angles (θ) from 1.3 to 51.8 and 47.8° at both sides of benzene rings and moderately modifies the Py-Ph, Py-NCO and NCO–NCO dihedral angles, Table 2. On the other hand, solvothermal reaction of **2** reversed the orientation of the pyridyl rings and recovered **1**.

#### 3.2.6. Emission Properties

The photoluminescence (PL) property of d^10^ metal complexes have potential applications as chemical sensors and fluorescent materials [24,25,26,27]. The solid state PL of **1**–**4**, and 1,4-pbpa were obtained at room temperature, Figure 5. The maximum excitation and emission wavelengths of **1** – **4**, and 1,4-pbpa are listed in Table 4, showing that the emissions of **1** – **4** and 1,4-pbpa appear at 448, 438, 455, 461, and 330 nm upon the excitations at 372, 367, 376, 396 and 273 nm, respectively. Due to the d^10^ electronic configuration of the Hg(II) metal ion that undergoes hardly either oxidation or reduction, the fluorescence emissions of **1**–**4** may result from the organic linkers and are probably attributable to π* → n or π* → π transitions [28]. The red shifts of the emission wavelengths for complexes **2**–**4** (Cl^−^ < Br^−^ < I^−^) may be ascribed to the different identity (electronegativity and size) of the halide anions, while coordination environment of the metal centers and arrangement of linkers also influence the luminescence [29].

## 4. Conclusions

Four Hg(II) halide CPs based on 1,4-pbpa have been synthesized and structurally characterized. While complex **1** forms a 1D sinusoidal chain, **2**–**4** are isostructural 1D helical chains. A comparison of the structures of the Hg(II) halide complexes involving 1,2-, 1,3-, and 1,4-pbpa ligands demonstrates that the co-crystallized solvent molecules and the ligand-isomerism of the spacer ligands play important roles in determining the structural diversity of the pbpa-based Hg(II) halide CPs, whereas the structure-directing role of the halide anion is not influential. Complex **1** undergoes reversible structural transformation with **2** upon removal and uptake of CH_3_OH, which represents the first example of HgCl_2_-containing CPs constructed from the bis-pyridyl-bis-amide ligands. The structural transformation can be ascribed to the pyridyl ring rotation of the 1,4-pbpa ligand that rearranges reversibly the ligand conformation between *trans* anti-anti and *trans* syn-anti. This study provides an insight into understanding the reversibility of the structural transformation invoked by the conformational change.

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
