# Peer review of "One-Dimensional Mercury Halide Coordination Polymers Based on A Semi-Rigid *N*-Donor Ligand: Reversible Structural Transformation"

_polymers, 2019, doi:10.3390/polym11030436_

Round 1

Reviewer 1 Report

The paper by Chhetri et al deals with the synthesis and X-ray (both single-crystal and powder diffraction studies) structural characterization of four coordination polymers assembled in solution by stratification of mercury(II) halides, HgX2 (X = Cl, Br, I) dissolved in methanol or ethanol, with the nitrogen-donor ligand 2,2'-(1,4-phenylene)-bis(N-(pyridin-3-yl)acetamide) (1,4-pbpa) dissolved in THF. The effect on the polymeric structure given by the variation of methanol with ethanol, as well as the modulation given by the halide anion, is well described and characterized. In particular, the reversible transformation of the synusoidal chain complex [Hg(1,4-pbpa)Cl2.CH3OH]n, 1, in the helical polymer [Hg(1,4-pbpa)Cl2]n, 2, and viceversa is reported and described through X-ray powder diffraction studies. A probable mechanism in the structural rearrangment from 1 to 2 due to the loss of the methanol molecule is given as well. 

I really appreciate reading the paper, which is well written and referenced. I suggest the publication on Polymers in the Special Issue dedicated to metallopolymers after two minor changes:

1 - a row with the temperature (in K) of the diffraction experiments, with the relative error, has to be added in Table 1;

2 - in the experimental section, I'd rather prefer to see selected infrared stretchings and bendings with the corresponding assignement, instead of the list of all observed bands.

Author Response

Reviewer: 1
The paper by Chhetri et al deals with the synthesis and X-ray (both single-crystal and powder diffraction studies) structural characterization of four coordination polymers assembled in solution by stratification of mercury(II) halides, HgX2 (X = Cl, Br, I) dissolved in methanol or ethanol, with the nitrogen-donor ligand 2,2'-(1,4-phenylene)-bis(N-(pyridin-3-yl)acetamide) (1,4-pbpa) dissolved in THF. The effect on the polymeric structure given by the variation of methanol with ethanol, as well as the modulation given by the halide anion, is well described and characterized. In particular, the reversible transformation of the synusoidal chain complex [Hg(1,4-pbpa)Cl2.CH3OH]n, 1, in the helical polymer [Hg(1,4-pbpa)Cl2]n, 2, and viceversa is reported and described through X-ray powder diffraction studies. A probable mechanism in the structural rearrangment from 1 to 2 due to the loss of the methanol molecule is given as well. 

I really appreciate reading the paper, which is well written and referenced. I suggest the publication on Polymers in the Special Issue dedicated to metallopolymers after two minor changes:

1 - a row with the temperature (in K) of the diffraction experiments, with the relative error, has to be added in Table 1;

Ans: A row with the temperature (in K) of the diffraction experiments, with the relative error, has been added to Table 1. Please see page 3.

2 - in the experimental section, I'd rather prefer to see selected infrared stretchings and bendings with the corresponding assignement, instead of the list of all observed bands.

Ans: Infrared stretchings and bendings for the amide groups of the ligands in 14 have been added to the experimental section. Please see pages 2 and 3.

Reviewer 2 Report

Paper of Chettri et al. describes four one-dimensional (1D) mercury(II) halide coordination polymers among which one undergoes structural phase transformation from sinusoidal structure into helical structure. This is definitely the biggest value in this manuscript. Crystal structures are described in detail and necessary characterizations have been made. I think that this manuscript fulfill all standards of Polymers journal and that will be a citable paper. Therefore my recommendation is minor revision - here I provide necessary corrections that should be made before publication.

-Some expresssions are not understandable. For example, "solvent removal and adoption" - do Authors have in mind by "adoption" the exposure to a vapor of given solvent (here acetonitrile)? Please correct throughout manuscript.

-In experimental section much more experimental details should be provided for all techniques reported. For example, please report emission spectroscopy excittion and emission slits, as it seems that compounds differed in emission intensity.

-The quality of Scheme 2 is disproportinately weak when compared to the other Figures and Schemes. It should be redrawn.

-Please have one more look on citations. For example in reference 9 the name of one of Authors should be Bauzá, not Bauz

Finally, I would like to suggest Authors the use of other techniques for investigation of phase transformations, e.g. DSC, or SHG (when applicable). Of course, compounds presented in this paper undergo phase transformations in harsh "wet" conditions, but I think for future, more multidisciplinarity can add up value to future studies.

Author Response

Reviewer: 2
Paper of Chettri et al. describes four one-dimensional (1D) mercury(II) halide coordination polymers among which one undergoes structural phase transformation from sinusoidal structure into helical structure. This is definitely the biggest value in this manuscript. Crystal structures are described in detail and necessary characterizations have been made. I think that this manuscript fulfill all standards of Polymers journal and that will be a citable paper. Therefore my recommendation is minor revision - here I provide necessary corrections that should be made before publication.

-Some expresssions are not understandable. For example, "solvent removal and adoption" - do Authors have in mind by "adoption" the exposure to a vapor of given solvent (here acetonitrile)? Please correct throughout manuscript.

Ans: We have replaced “adoption” with “uptake” and used “removal and uptake of CH3OH”. Please see pages 1, 2 and 8.

-In experimental section much more experimental details should be provided for all techniques reported. For example, please report emission spectroscopy excittion and emission slits, as it seems that compounds differed in emission intensity.

Ans: we have added the excitation slit and emission slits to experimental section please see page 2.

-The quality of Scheme 2 is disproportinately weak when compared to the other Figures and Schemes. It should be redrawn.

Ans: We have enlarged Scheme 2 and it looks much better.

-Please have one more look on citations. For example in reference 9 the name of one of Authors should be Bauzá, not Bauz

Ans: We have corrected the mistakes. Please see references 9 and 28.

Finally, I would like to suggest Authors the use of other techniques for investigation of phase transformations, e.g. DSC, or SHG (when applicable). Of course, compounds presented in this paper undergo phase transformations in harsh "wet" conditions, but I think for future, more multidisciplinarity can add up value to future studies.

Ans: We thank the reviewer’s suggestion. Where ever it is applicable, we will consider these techniques in the future.
